# Effect of Ginseng Extracts on the Improvement of Osteopathic and Arthritis Symptoms in Women with Osteopenia: A Randomized, Double-Blind, Placebo-Controlled Clinical Trial

**DOI:** 10.3390/nu13103352

**Published:** 2021-09-24

**Authors:** Su-Jin Jung, Mi-Ra Oh, Dae Young Lee, Young-Seob Lee, Geum-Soog Kim, Soo-Hyun Park, Soog-Kyoung Han, Young-Ock Kim, Sun-Jung Yoon, Soo-Wan Chae

**Affiliations:** 1Clinical Trial Center for Functional Foods, Jeonbuk National University Hospital, Jeonju 54907, Jeonbuk, Korea; sjjeong@jbctc.org (S.-J.J.); mroh@jbctc.org (M.-R.O.); 2Biomedical Research Institute, Jeonbuk National University Hospital, Jeonju 54907, Jeonbuk, Korea; 3Department of Herbal Crop Research, National Institute of Horticultural and Herbal Science, Rural Development Administration (RDA), Eumseong 27709, Chungbuk, Korea; dylee0809@gmail.com (D.Y.L.); youngseoblee@korea.kr (Y.-S.L.); kimgs0725@korea.kr (G.-S.K.); kyo9128@korea.kr (Y.-O.K.); 4Korea Food Research Institute, Wanju 55365, Jeonbuk, Korea; shpark0204@kfri.re.kr; 5Department of Food Science and Human Nutrition, Jeonbuk National University, 567 Baekje-daero, Jeonju 54896, Jeonbuk, Korea; skhan27@hanmail.net; 6Department of Orthopedic Surgery, Medical School, Jeonbuk National University, 567 Baekje-daero, Jeonju 54896, Jeonbuk, Korea; sunjungyoon@jbnu.ac.kr

**Keywords:** ginseng extract, bone metabolism, osteopenia, arthritis symptoms, WOMAC

## Abstract

Ginsenosides are active compounds that are beneficial to bone metabolism and have anti-osteoporosis properties. However, very few clinical investigations have investigated the effect of ginseng extract (GE) on bone metabolism. This study aims to determine the effect of GE on improving bone metabolism and arthritis symptoms in postmenopausal women with osteopenia. A 12-week randomized, double-blind, placebo-controlled clinical trial was conducted. A total of 90 subjects were randomly divided into a placebo group, GE 1 g group, and GE 3 g group for 12 weeks based on the random 1:1:1 assignment to these three groups. The primary outcome is represented by bone metabolism indices consisting of serum osteocalcin (OC), urine deoxypyridinoline (DPD), and DPD/OC measurements. Secondary outcomes were serum CTX, NTX, Ca, P, BsALP, P1NP, OC/CTX ratio, and WOMAC index. The GE 3 g group had a significantly increased serum OC concentration. Similarly, the GE 3 g group showed a significant decrease in the DPD/OC ratio, representing bone resorption and bone formation. Moreover, among all the groups, the GE 3 g group demonstrated appreciable improvements in the WOMAC index scores. In women with osteopenia, intake of 3 g of GE per day over 12 weeks notably improved the knee arthritis symptoms with improvements in the OC concentration and ratios of bone formation indices like DPD/OC.

## 1. Introduction

Osteoporosis is a metabolic bone disease caused by the collapse of the balance between bone formation and bone resorption. It progresses with age, especially in women after menopause. In the developed world, the incidence of osteoporosis and its treatment is increasing with the aging population. In Korea, the progressive increase in the elderly population created a social and medical interest in osteoporosis. Specifically, after menopause, women experience 20~30% bone loss in their sponge bones due to reduced estrogen hormone secretion, thereby increasing the risk of spinal compression fractures and osteoporosis as bone resorption exceeds bone formation [1,2,3,4,5]. In people over 50 years of age, the early diagnosis and prevention of bone loss are more important than the treatment of osteoporosis developed after fractures since the rapid aging trend potentially multiplies the risk of osteoporosis [6,7,8,9]. The cause of osteoporosis is unclear, and factors, such as endocrinology, nutrition, and genetic factors, contribute to the development of osteoporosis. Moreover, deficiency of calcium, estrogen, and vitamin D, and excess phosphorus and protein enhances the development of osteoporosis [10,11]. 

About 5% of the elderly population suffer from osteoporosis in developed countries and poses a major risk to postmenopausal women. The use of vitamins, calcium, and natural products prevents osteoporosis in the elderly population. Supplementation of female hormones is suggested in postmenopausal women to improve osteoporosis [12,13,14,15,16,17,18]. However, there is practically no cure for osteoporosis, and the efficacy of supplements is not guaranteed either. Thus, investigations focus principally on natural food products that improve osteoporosis with fewer side effects. Ginseng (*Panax ginseng* C.A. Meyer) is one of the most well-known medicinal plants worldwide and is a representative herbal medicine in Korea, with over 2000 years of medicinal history [19]. Traditionally, it has been used in Asia to maintain body homeostasis and strengthen bioenergy. The key active substances in ginseng include ginsenosides, polyacetylenes, phenolic compounds, acidic polysaccharides, peptides, and alkaloids [20]. Among these, ginsenosides are the important pharmacologically active ingredients of ginseng. Previously, ginseng has been known for treating cancer [21] and acute respiratory illness [22], improving blood circulation [23,24], controlling fatigue and stress [25,26], nervous system protection [27,28], treating diabetes [29], exercise performance improvement [30], and cognitive [31] and liver function improvement [32]. Panax notoginseng saponins (PNSs) are reported as a potential alternative medicine for the prevention and treatment of postmenopausal osteoporosis [33,34]. However, most of the reports are non-clinical and lack sufficient evidence to determine the impact of a single ginseng extract (GE) on bone metabolism. Recent preclinical investigations (in vitro, in vivo) have demonstrated that GE has a beneficial effect on improving the bone metabolism index and osteoporosis along with improvements in bone biomarkers, including bone density and bone mass [35,36,37]. The clinical trials evaluating the impact and relevance of GE supplementation in improving bone metabolism are very limited. Thus, a randomized, double-blind, placebo-controlled was conducted to assess the effectiveness and safety of GE supplementation on bone metabolism and arthritis symptoms in women with osteopenia.

## 2. Materials and Methods

### 2.1. Test Supplements

Test products given to the subjects were provided by the National Institute of Horticultural & Herbal Science, Rural Development Administration (Eumseong, Chungbuk, Korea). For this study, we obtained the same GE that was used for the analysis and preclinical investigations. To prepare the GE, fresh *Panax ginseng* (4 years old) was extracted with hot water under reflux (80 ± 5 ℃) for 8 h and filtered through a 5 μm filter. The supernatant was vacuum concentrated under reduced pressure (600 ± 100 mmHg, below 65 ℃) to attain 15 Brix materials. Finally, the extract of *P. ginseng* (referred to as GE) was freeze-dried under reduced pressure (−45 °C, 100 mTorr) for 48 h. For bulk production, the GE manufacturing process was optimized and analyzed using HPLC [38]. GE yield was 8.61%, and the sum of ginsenoside Rg1 and Rb1 was 8.61 mg/g. The representative HPLC chromatogram for GE is presented in Figure 1. Previous studies have shown that supplementation of GE at 300 mg/kg markedly improved the serum osteocalcin (OC) and calcium content in aged rats, resulting in reduced bone loss [37]. Thus, test products given in this study were prepared accordingly. The Ministry of Food and Drug Safety, Korea has indicated 0.5~5 g of ginseng powder or 3~80 mg of ginsenoside for daily intake. Considering the limitations and outcomes of the previous studies, 1 (low dose) and 3 g (high dose) of GE were used for this study. The test product (GE) was provided in a capsule form (dry yellow powder), and the placebo product was prepared with cellulose crystals and had the same appearance, aroma, and weight as the test product (Table 1).

### 2.2. Participants

This study was conducted upon the approval from the IRB (Institutional review board IRB), Jeonbuk National University Hospital (JUH IRB number 2015-06-3). The entire process of this clinical trial was carried out in accordance with the Helsinki Declaration and the provision of KGCP. The protocol was registered at www.clinicaltrials.gov (NCT02763280). The participants for the study were recruited through floor advertising (brochure, poster, Jeonbuk National University Hospital Web page). The selected participants were provided with an outline of the trial and agreed to participate in the study. Volunteers were screened according to the inclusion and exclusion criteria, and their written informed consent was obtained. The participants for this trial were selected after the following criteria were met: (1) A woman aged more than 40 years at the time of the screening test, for whom six months has passed after the period of amenorrhea; (2) a person with a T-score (bone density test, DEXA) of −1.0 or less; (3) a person whose blood OC concentration, which is a blood formation index, is greater than 8 ng/mL and whose urine deoxypyridinoline (DPD) concentration, which is a blood absorption index, is greater than 5.2 nM DPD/mL creatinine; and (4) participants who listened to the detailed explanations of this study, fully understood, and agreed in writing to participate and comply with precautions voluntarily. Participants were excluded from this study if the subjects had any of the following criteria: (1) a person diagnosed with secondary osteopenia or osteoporosis; (2) a person who was on female hormone replacement therapy, calcitonin, strontium ranelate, and bisphosphonate (alendronate, risedronate, ibandronate, and zoledronate) to treat osteoporosis within six months prior to the screening test; (3) a person taking vitamin D, K, and Ca supplements within four weeks prior to the screening test; (4) a person taking beta-adrenaline receptor type inhibitors within two months prior to the screening test; (5) a person who had been continuously taking drugs and healthy functional foods that affect the interpretation of the study results related to hyperlipidemia or endocrine diseases, hormone preparations, and bone health improvement within two months prior to the screening test; (6) cancer patients with breast cancer, ovarian cancer, endometrial cancer, cervical cancer, bladder cancer, etc.; (7) a person who had a disease affecting the bone metabolism (thyroid disease, diabetes, liver disease, kidney disease, etc.); (8) a person who had osteoporosis fractures, degenerative spinal disease, femur disease, hip disease, etc., or had a history of these diseases; (9) those whose body mass index (BMI) was less than or equal to 18.5 kg/m^2^ or greater than 30 kg/m^2^; (10) a person suspected of cognitive impairment; (11) a person who is under medication and has clinically significant hypersensitivity reactions (however, subjects could participate in this study in spite of their condition at the discretion of the person in charge of this study); (12) a person with a history of gastrointestinal diseases (e.g., Crohn’s disease) or gastrointestinal surgery (excluding simple appendectomy or hernia surgery) that could affect the absorption of the test products used; (13) a person who had taken antipsychotic medication within two months prior to the screening test; (14) a person with a history of drug or alcohol abuse; (15) a person who had the following results in the diagnostic medical examination (AST, ALT > twice the upper limit of the reference, serum creatinine > 2.0 mg/dL); (16) a person who had participated in any other human study within two months prior to this screening test; and (17) a person who was deemed inappropriate to participate in the study based on the judgment of the person in charge of this study due to the results of the diagnostic medical examination and/or other reasons. 

### 2.3. Study Design 

This randomized, double-blind, placebo-controlled human study was conducted for 12 weeks, from 7 August 2015 to 24 June 2016. A total of 90 subjects were randomly assigned to three groups: a GE 1 g group (low), a GE 3 g group (high), and a placebo group at a rate of 1:1:1. Additionally, a block random assignment method was applied to reduce the predictability of the assigned groups. The selected participants underwent a series of tests, such as hematological tests, electrocardiogram tests, and bone density tests. Later, the participants who met the inclusion and exclusion criteria were subjected to a dual-energy X-ray absorptiometry (DEXA; Discovery W, Hologic, Marlborough, MA, USA) scan to determine the bone mass density (BMD). BMD measurements were conducted by scanning the lumbar spine and femur, with the subjects wearing a gown and lying in a comfortable position. The measurement results were rounded from the second decimal place to the first decimal place. The body measurements were used as input for the DEXA device. The participants in this study consumed 6 capsules/day: three capsules in the morning before breakfast and three capsules in the evening before dinner. The GE 1 g group took 1 g/day of GE, and the GE 3 g group had 3 g/day of GE. Subjects were monitored as required, and a total of three tests were performed to measure the efficacy and safety of the test product on bone metabolism. The screening was the first test. The second test was done on the first day (week 0), and the third test was done at the end of the 12th week. All the participants were given new test products and were asked to return the remaining test products from the last six weeks on their second visit (6th week). All the subjects were instructed to maintain their regular physical activity and food habits and asked not to consume any other functional foods or dietary supplements during the study period.

### 2.4. Outcome Measurements

A series of tests were done to verify the safety and efficacy of the test products at predetermined time points (baseline, week 0, and week 12). The blood samples of the subjects were collected on an empty stomach for more than 12 h from the day before, centrifuged at 3000 rpm for 20 min (Hanil Science Industrial Co. Ltd., Seoul, Korea), and frozen at −80 °C until the analysis. 

#### 2.4.1. Primary Outcomes

Changes in serum OC, urinary DPD, and DPD/OC (bone resorption/bone formation) ratio were set as the primary validity evaluation index for bone turnover markers.

Serum OC 

Serum OC levels reflecting the bone formation were determined by an immunoradiometric assay using an OSCA test osteocalcin kit (Brahms, Germany) [39]. 

Urinary DPD 

DPD reflecting systemic bone resorption was measured using an early morning urine specimen free of dietary influences and changes due to physical activity. DPD levels were determined using radioimmunoassay using a DPD RIA Kit (IDS, England), and the results reported were corrected for urine creatine. 

#### 2.4.2. Secondary Outcomes

The second validity evaluation refers to the bone resorption index in the blood bone metabolism index in which serum C-terminal telopeptide (CTX) and N-terminal telopeptide (NTX) were measured. The bone formation index was analyzed by serum BsALP (bone-specific alkaline phosphatase) and serum P1NP (procollagen type 1 n-terminal propeptide), and changes in the OC/CTX ratio, blood Ca, and phosphorus were measured and analyzed. The evaluation of the WOMAC (Western Ontario and McMaster Universities Arthritis) index, which is a degenerative arthritis index for the knee joint and a septic arthritis index for the hip, was performed using the WOMAC-Osteoarthritis index developed by Bellamy [40]. It consists of 24 items, which are classified into three sub-regions (pain, stiffness, and physical function). Each of these sub regions consists of 5 items (pain), 2 items (stiffness), and 17 items (physical function). The test questions are scored on a scale of 0–4, which corresponds to none (0), mild (1), moderate (2), severe (3), and extreme (4). The measurement scores for each sub-region vary: pain (0–20 points), stiffness (0~8 points), and physical function (0~68 points). The sum of the scores for all three sub-regions gives a total WOMAC score ranging from 0 to 96. Higher scores indicate severe physical dysfunction. 

### 2.5. Safety Measurements

To investigate the possible adverse events, the clinical conditions of the subjects were assessed before and during the trial on predetermined visits. The subjects’ vital signs were recorded as a safety assessment, and general hematological and chemical tests were performed. General hematological parameters like WBC, RBC, Hb, Hct, and platelet count were analyzed. Biochemical tests were done to determine the total bilirubin, alkaline phosphatase (ALP), gamma-glutamyl transferase (GGT), alanine aminotransferase (ALT), aspartate transaminase (AST), total cholesterol, triglyceride, HDL-C, LDL-C, glucose, total protein, albumin, BUN, creatinine, creatine kinase, and LDH in the blood. Urine samples were tested for specific gravity, pH, WBC, nitrite, protein, glucose, ketone, bilirubin, urobilinogen, and occult blood, and hormone tests were performed by measuring E2 and FSH. 

### 2.6. Evaluation of Diet and Physical Activity

The subjects were asked to record the dietary intakes for 3 days (two days of the week, one day of the weekend) before the first visit (week 0) and the third visit (week 12). Based on these recorded and retrieved diet records, average daily dietary intake was analyzed using Can-Pro 4.0 software (The Korean Nutrition Society, Seoul, Korea) and presented as average values. Physical activity was evaluated according to a metabolic equivalent task (MET) assessment using the global physical activity questionnaire (GPAQ) [41]. 

### 2.7. Sample Size

The primary outcome of this study is a change in the serum OC between the GE and placebo groups after 12 weeks of ingestion. Power calculation was implemented based on the results of a previous study [11]. The analysis of the main evaluation parameter (OC) in the sample size calculation of this study was based on the one-way ANOVA method in which the changes in OC after 12 weeks of intake in the GE 3 g (high) group, GE 1 g (low) group, and the placebo group were assumed as 4.75, 3.7, and 2.8 ng/mL, respectively, and SD was assumed as 2.2 ng/mL. The number of subjects needed in each group to achieve 80% statistical power for a 5% significance level (two-sided test) was 24 subjects per group. Thus, 90 subjects for three groups were enrolled, considering a dropout ratio of about 20%.

### 2.8. Statistical Analysis

The statistical analyses were performed using the SAS version 9.2 (SAS Institute, Cary, NC, USA). Continuous variables and frequencies are presented as mean ± SD. The results of the study were analyzed with the intention-to-treat (ITT) method. Categorical variables were analyzed with the chi-square test (Fisher’s effect test), and an independent *t*-test was used to determine the mean comparison between the two groups. Further, one-way ANOVA was used to analyze the homogeneity. The paired *t*-test was used to analyze the difference between the before and after tests. Similarly, the difference between the intake groups and within the intake group was analyzed using a linear mixed model. Additionally, homogeneous items were treated as covariates and analyzed with repeated measures ANCOVA. The statistical significance level was set as *p* < 0.05. 

## 3. Results

### 3.1. Demographic Characteristics of Participants

The general characteristics of the subjects in this study are presented in Table 2. The age, menopause age, menopause period, birth count, BMI, vital signs, smoking status, spinal bone density, femoral bone density, osteocalcin, DPD, and CTX of the participants were recorded as a baseline index. There were no significant differences (*p* > 0.05) between the groups. A total of 133 subjects participated in the screening test, from which a total of 90 subjects were selected depending on the exclusion and inclusion criteria. The subjects selected for the study were randomly divided into three groups, and each group consisted of 30 participants. Four of the 90 registered subjects withdrew their consent (one subject in the GE 1 g group and three in the placebo group), and one subject in the GE 1 g group failed to comply with the test product. Together, a total of 85 subjects (28 in the GE 1 g group, 30 in the GE 3 g group, and 27 in the placebo group) completed all the specified procedures. Final data were presented using data from 85 subjects (Figure 2). The compliance with the test products in the GE 1 g group, GE 3 g group, and placebo group was 94.0 ± 5.8%, 95.2 ± 5.0%, and 90.10 ± 8.9%, respectively. There were no significant differences between groups (*p* > 0.05). In the post-hoc test results, the compliance with the ingested product of the placebo group was significantly lower than the GE 3 g group.

### 3.2. Diet Intake and Physical Activity

The nutrient intake of the participants in different groups during the 12-week intervention period is presented in Table 3. There were no significant differences between the groups’ daily intake of calories, carbohydrates, proteins, fats, and fibers (*p* > 0.05) before and after the completion of the intervention period. Similarly, there were no significant differences between the group’s intake of vitamin D, vitamin C, Ca, and P affecting calcium metabolism and bone health. The degree of physical activity affects the bone metabolism [11] and possibly influences the study outcomes. In this study, metabolic equivalents (MET) analysis reveals that physical activity in the placebo group after 12 weeks significantly increased compared to the baseline physical activity. However, there were no significant differences between the three groups. Still, considering the placebo group’s increased physical activity, further analysis was implemented by calibrating it with MET values during the analysis of the validation parameters.

### 3.3. Efficacy Evaluation

#### 3.3.1. Primary Outcome

In this study, changes in the serum OC and DPD/OC ratio of the subjects in the GE 3 g group were significantly different. However, changes in the serum OC and DPD/OC ratio of the subjects in the placebo and GE 1 g group were less than the GE 3 g group (Table 4 and Figure 3). Moreover, changes in DPD in all the groups were not significant. Further, the results of the ANCOVA (after calibration with MET values) GE 3 g group (high) showed an increase in the concentration of serum OC compared to the GE 1 g group (low) and placebo group while the DPD/OC ratio decreased.

#### 3.3.2. Secondary Outcome

The changes in the bone turnover markers of the study are presented in Table 4. The calcium concentration in the GE 3 g group significantly increased during the intervention period compared to the GE 1 g (low) and placebo groups (*p* < 0.05). Similarly, the OC/CTX ratio remarkably increased in the GE 3 g group compared to the GE 1 g (low) and placebo groups. Further, the results of ANCOVA (after calibration with MET values) showed notable differences between the groups as the GE 3 g group showed an increased blood Ca concentration and OC/CTX ratio (*p* < 0.05). The post-hoc analysis shows a decreased blood Ca concentration and OC/CTX ratio in the GE 1 g group (low). However, there were no significant differences in the levels of the bone formation indices, such as serum CTX, NTX, BsALP, and P1NP, in all three groups. The changes in WOMAC scores are listed in Table 5. The pain Q3 (knee pains disrupting sleep at night) decreased significantly in the GE 3 g group compared to the placebo group after 12 weeks of intervention. Additionally, there were significant differences in the stiffness of joints (Q2: stiffness of joints after sitting, lying, or resting in the afternoon) between the three groups (*p* < 0.05), and the GE 3 g group (high) showed a significant decrease in this level compared to the GE 1 g (low) and placebo groups. 

Moreover, the results of ANCOVA (after calibration with MET values) showed significant differences in the stiffness of joints (Q2) between the groups (*p* < 0.05). Among the three groups, the GE 3 g group (high) demonstrated a significant decrease in the stiffness of joints (Q2) value compared to the GE 1 g (low) and placebo groups. Further, the post-hoc analysis revealed that pain (Q3) was improved in the GE 3 g group (high) compared to the placebo group. Similarly, stiffness (Q2) was improved in the GE 3 g group (high) compared to the GE 1 g group (low). Collectively, there were significant differences between the three groups (*p* < 0.05). The WOMAC index score in the GE 3 g group (high) was significantly decreased compared to in the GE 1 g (low) and placebo groups. However, there were no significant differences between the three groups. 

### 3.4. Safety and Adverse Events

During the intervention period, a total of 16 mild to moderate adverse events occurred in 14 subjects out of a total of 90 subjects. However, there was no difference in the occurrence of adverse events between the groups. The adverse events during the study included one rash, nine colds, one headache, one indigestion, one skin rash, one abdominal discomfort, one dizziness, and one indigestion. Analysis of the adverse events revealed no immediate relation between the adverse reactions and ingestion of the test product. Vital signs and biochemical tests (hematological, blood biochemistry, and urine) were within the normal range (Appendix A). Hence, the results of these tests had no clinical significance. 

## 4. Discussion

This study determined that the postmenopausal women group with osteopenia that consumed GE 3 g (high) a day for 12 weeks showed a significant increase in bone formation indices, serum OC, and blood Ca level as compared to the placebo group. No clinically meaningful adverse events or changes in the body were observed during this study, and thus the intake of GE was judged to be safe for the human body. This study is the first randomized, double-blind, placebo-controlled clinical trial to determine the effect of GE on improving bone metabolism and arthritis symptoms in postmenopausal women with osteopenia. In this study, intake of GE 3 g a day for 12 weeks in postmenopausal women with osteopenia showed improved bone formation indices like serum OC and blood Ca levels compared to the placebo group. Additionally, the absence of adverse reactions suggests the safety of GE during the study. The levels of bone resorption indices and biomarkers like serum OC, BsALP, urine DPD, serum NTX, and CTX are crucial to bone remodeling [42]. However, measurement of some of these indices does not exactly associate with improved bone metabolism. OC is one of the bone formation indices, and it is produced by osteoblasts and is accumulated in the extracellular matrix of the bone. About 30% of the newly synthesized OC is released into the blood, reflecting improved bone formation. However, the synthesis of OC alone does not predict bone quality and quantity. OC works well in combination with calcium to form bone crystals, potentially contributing to bone quality and quantity. Thus, an increase in OC is dependent on calcium intake [14,43,44]. Additionally, urine DPD, a specific marker that reflects the degree of bone resorption, has the advantage of being impervious to food and is largely used as a bone resorption index [45]. Previous in vitro and in vivo investigations have identified the mechanism by which GE regulates the bone reformation process [35,36,37]. Data from these studies suggest that the bioactive compound ginsenoside Re in the GE stimulates the osteoblasts by activating osteoblast markers like runt-related transcription factor, type 1 collagen, ALP, and OC in the mouse osteoblastic cell line MC3T3-E1. Additionally, ginsenoside Re inhibits the differentiation of osteoclasts [35,36]. Moreover, an increased bone mineral concentration significantly improves bone-related biomarkers, including femur bone density and bone mass. The increase in serum OC and blood Ca levels in aged animal models confirms GE’s beneficial effect on bone metabolism. Here, supplementation of GE 3 g per day significantly increased the OC and blood Ca concentration compared to the placebo group. An increase in OC can effectively inhibit greater bone turnover rates and bone loss, improving age-linked osteoporosis. Additionally, the DPD and DPD/OC ratios are important as bone resorption indices [46]. The DPD/OC ratio provides information similar to the bone density evaluated by individual skeletal X-ray of the body and is strongly associated with bone diseases. The DPD/OC ratio increases significantly in patients with bone diseases compared to those without the disease [46]. In general, nutritional intake is recognized as a key factor for bone health. Dietary factors like calcium, phosphorus, vitamin D, and protein influence bone density in young adults [47]. In middle-aged women, calcium supplements have been reported to contribute to significant increases in blood OC and bone density [15,16,17]. In this study, dietary intake analysis revealed that vitamin D, calcium, and protein intake during participation were similar the in two groups. Thus, it can be presumed that these dietary intakes did not influence the outcome of the study. Analysis ANCOVA (after calibration with MET values) confirmed that the improvements in the bone metabolism index in the GE 3 g group are consistent and significant compared to the placebo group. Recent investigation demonstrated that supplementation of Oligopin^®^ (maritime pine bark extract 150 mg/day) over 12 weeks in postmenopausal women with osteopenia significantly improved bone metabolism as the serum OC and OC/CTX-1 ratio increased. These outcomes were similar to the outcomes of this study. 

In this study, GE 3 g (high) supplementation for 12 weeks effectively increased the serum OC level without significant changes in DPD. The DPD/OC ratio is assumed to have a positive effect on bone metabolism as it is tilted towards bone formation. Previous studies have shown that the serum OC level in postmenopausal women and other key factors reflecting bone metabolism can be considered as independent predictors of metabolic disease (MD) in premenopausal and postmenopausal women. Another study investigating the association between serum osteocalcin and metabolic syndrome (MetS) in premenopausal and postmenopausal women revealed that women without MetS have higher serum OC than women with MetS. Serum osteocalcin levels decreased with the increasing number of MetS elements [48]. Ginsenoside, the active substance in ginseng, is known for its anti-osteoporotic effects. In an aqueous ginseng extract, ginsenoside Re is the major component. In an in vivo zebrafish model, aqueous ginseng extract and ginsenoside Re were demonstrated to have an anti-osteoclastogenic effect [35]. Ginsenosides Rd and Re are major bioactive compounds in GE. In this study, serum OC and Ca concentrations increased with supplementation of GE. Thus, the reduced DPD/OC ratio suggests the potential application of GE in preventing bone-related disorders. Additionally, supplementation of GE improved the WOMAC index. Specifically, intake of GE 3g (high) per day for 12 weeks significantly decreased the sleep-disturbing symptoms, such as knee joint pain and stiffness. Previous reports have revealed that decreased serum OC is associated with an increased total WOMAC score in patients with knee osteoarthritis [49]. Consistent with these observations, the current study outcomes showed that GE supplementation improved the serum OC and reduced the total WOMAC score, indicating improved joint pain and stiffness caused by osteopenia. The study has some limitations. Primarily, the results of this study do not explain the definitive effect of GE on bone metabolism. Additionally, the study period of 12 weeks appears to be short and is not enough to evaluate the natural test product. Generally, natural food or functional foods take an extended period to show their potential therapeutic benefits. Hence, the results of this study cannot be generalized based on the results of this study alone. Thus, it is necessary to conduct extensive investigations by increasing the tracking period by more than one year with a significant number of subjects. However, the major results of the study are consistent with previous in vitro and in vivo observations. 

## 5. Conclusions

In summary, 12-week GE supplementation in postmenopausal women with osteopenia resulted in improved key indices of bone formation like serum OC concentrations and DPD/OC ratio. Collectively, the study confirms the beneficial effect of GE on bone metabolism. 

## Figures and Tables

**Figure 1 nutrients-13-03352-f001:**
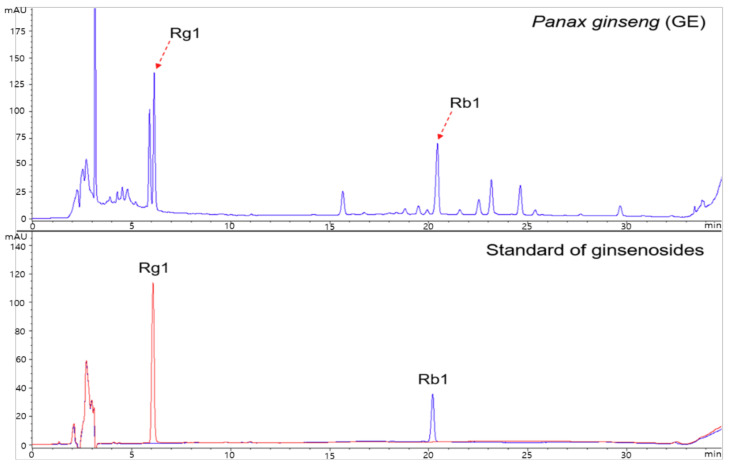
HPLC chromatogram of the freeze-dried hot-water extracts (GE) from *Panax ginseng* (4 years old).

**Figure 2 nutrients-13-03352-f002:**
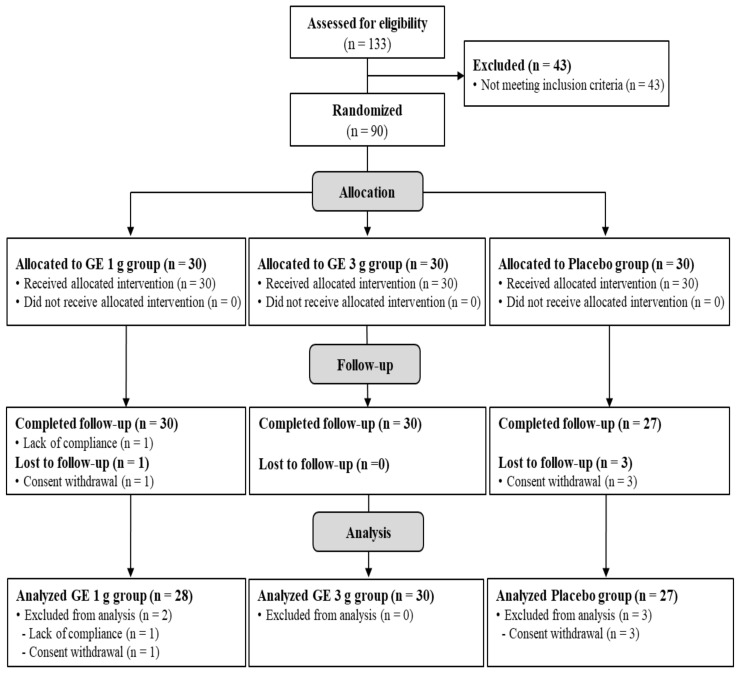
Flow diagram showing the subject recruitment process in the study.

**Figure 3 nutrients-13-03352-f003:**
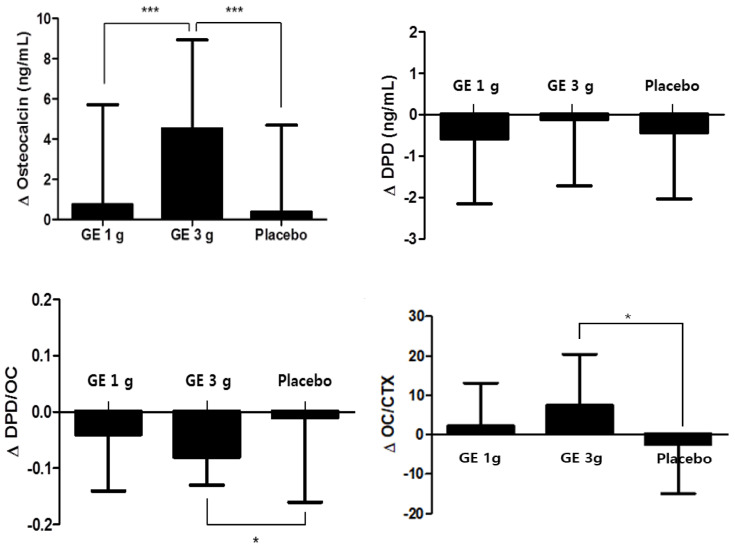
Changes in OC, DPD, DPD/OC ratio, and OC/CTX ratio of the subjects. There was a difference in the changes of the OC and DPD levels between the GE 1 g (low dose) and placebo groups. The DPD/OC and OC/CTX ratio was significantly decreased in the GE 3 g group (high dose) after 12 weeks. * *p* < 0.05, *** *p* < 0.001. Abbreviations: OC, Osteocalcin; DPD, deoxypyridinoline; CTX, C-terminal telopeptide.

**Table 1 nutrients-13-03352-t001:** Composition of the test and placebo products.

Ingredients	Contents (%)		
	Ginseng Extract Powder 1 g (Low Dose)	Ginseng Extract Powder 3 g (High Dose)	Placebo
Ginseng powder	33.3	>99.0	
Silicon dioxide	0.50	0.5	
Magnesium stearate	0.24	0.5	0.37
Microcrystalline cellulose	65.53		98.98
Ginseng flavor	0.397		0.60
Caramel coloring	0.026		0.04
Gardenia yellow coloring	0.007		0.01
Total	100	100	100

**Table 2 nutrients-13-03352-t002:** Demographic characteristics of the study participants.

Variables	GE Low (1 g) Group (*n* = 30)	GE High (3 g) Group (*n* = 30)	Placebo Group (*n* = 30)	Total (*n* = 90)	*p*-Value ^1^
Age, years	55.40 ± 3.17	53.93 ± 2.82	54.63 ± 3.02	54.66 ± 3.03	0.174
Menopause age (years)	50.27 ± 3.94	48.57 ± 4.04	49.43 ± 4.01	49.42 ± 4.01	0.263
Menopause duration (months)	61.87 ± 40.05	64.00 ± 55.78	62.40 ± 50.76	62.76 ± 48.75	0.985
Number of pregnancies	2.07 ± 0.58	2.20 ± 0.81	2.23 ± 0.86	2.17 ± 0.75	0.668
Height (cm)	156.57 ± 4.25	155.20 ± 6.22	155.37 ± 5.50	155.71 ± 5.36	0.565
Weight (kg)	58.45 ± 7.80	59.52 ± 9.31	57.21 ± 7.97	58.39 ± 8.35	0.569
Body mass index (kg/m^2^)	23.85 ± 3.01	24.65 ± 3.03	23.66 ± 2.69	24.05 ± 2.91	0.379
Drinking (yes/no)	10/20	9/21	12/18	31/59	0.709 ^2^
Alcohol consumption (units/week)	1.14 ± 1.18	1.10 ± 1.79	2.31 ± 2.68	1.58 ± 2.06	0.305
Current, smoker (yes/no)	1.14 ± 1.18	1.10 ± 1.79	2.31 ± 2.68	1.58 ± 2.06	0.305
VBD(T-score)	−2.25 ± 0.75	−1.99 ± 0.64	−1.96 ± 0.71	−2.07 ± 0.71	0.207
FBD(T-score)	−1.40 ± 0.88	−1.15 ± 0.67	−1.22 ± 0.77	−1.26 ± 0.78	0.433
OC (ng/mL)	21.3 ± 5.72	18.93 ± 3.72	21.11 ± 6.34	20.4 ± 5.38	0.173
DPD (nM DPD/mM crea)	7.43 ± 1.33	7.33 ± 1.52	8.29 ± 4.74	7.67 ± 2.91	0.410
DPD/OC ratio	0.37 ± 0.1	0.40 ± 0.09	0.42 ± 0.26	0.39 ± 0.17	0.538
CTX (ng/mL)	0.48 ± 0.11	0.49 ± 0.16	0.45 ± 0.16	0.48 ± 0.15	0.534
OC/CTX ratio	44.48 ± 11.39	41.10 ± 12.16	49.92 ± 15.51	45.16 ± 13.02	0.078
NTX (ng/mL)	15.58 ± 2.88	15.95 ± 4.05	15.93 ± 4.03	15.82 ± 3.66	0.914
Ca (mg/dL)	9.45 ± 0.36	9.35 ± 0.27	9.33 ± 0.25	9.38 ± 0.3	0.260
P (mg/dL)	3.67 ± 0.35	3.81 ± 0.33	3.72 ± 0.49	3.74 ± 0.39	0.421
BsALP (μg/L)	16.59 ± 5.06	15.91 ± 3.56	17.23 ± 5.93	16.55 ± 4.88	0.601
P1 NP (pg/mL)	61.23 ± 13.69	59.10 ± 14.97	58.18 ± 15.98	59.51 ± 14.78	0.737
WOMAC index	10.54 ± 12.75	14.7 ± 12.63	11.2 ± 11.87	8 ± 9.12	0.096

Values are presented as mean ± SD or number. ^1^ Analyzed by one-way ANOVA ^2^ Analyzed by Fisher’s exact test. GE low group: ginseng powder 1 g group, GE high group: ginseng powder 3 g group. Abbreviations: VBD, vertebral bone density; FBD, femoral bone density; OC, Osteocalcin; DPD, deoxypyridinoline; CTX, C-terminal telopeptide, NTX, N-terminal telopeptide, BsALP, bone specific alkaline phosphatase, P1 NP, procollagen type 1 n-terminal propeptide, WOMAC, western Ontario and McMaster Universities Arthritis.

**Table 3 nutrients-13-03352-t003:** Nutrient intakes of the study subjects during the 12-week intervention period.

	GE Low (1 g) Group (*n* = 28)				GE High (3 g) Group (*n* = 30)				Placebo Group (*n* = 27)				
	0 Week	12 Week	Diff	*p*-Value ^1^	0 Week	12 Week	Diff	*p*-Value ^1^	0 Week	12 Week	Diff	*p*-Value ^1^	*p*-Value ^2^
Calorie (kcal)	1681.30 ± 434.63	1609.45 ± 464.11	−71.85 ± 508.34	0.461	1522.94 ± 495.53	1557.56 ± 354.22	34.62 ± 502.06	0.708	1584.74 ± 426.59	1478.53 ± 409.78	−106.21 ± 416.01	0.196	0.510
Carbohydrate (g)	267.60 ± 78.58	267.70 ± 78.98	0.10 ± 90.58	0.995	244.00 ± 73.47	250.91 ± 62.14	6.90 ± 80.63	0.643	257.09 ± 64.63	246.56 ± 57.16	−10.54 ± 66.31	0.417	0.712
Lipid (g)	40.34 ± 20.35	34.78 ± 16.60	−5.56 ± 18.90	0.131	36.26 ± 20.88	38.64 ± 20.24	2.38 ± 28.35	0.649	39.90 ± 23.55	33.58 ± 18.65	−6.31 ± 25.34	0.207	0.333
Protein (g)	67.21 ± 21.18	63.08 ± 23.21	−4.13 ± 26.26	0.412	61.62 ± 27.87	60.28 ± 21.67	−1.34 ± 25.26	0.774	54.70 ± 20.10	55.33 ± 20.06	0.63 ± 22.62	0.886	0.774
Fiber (g)	23.13 ± 7.98	24.94 ± 8.54	1.81 ± 11.71	0.421	22.26 ± 11.93	23.14 ± 9.14	0.89 ± 10.08	0.634	21.89 ± 7.00	21.17 ± 5.86	−0.72 ± 7.40	0.616	0.635
Vitamin D (μg)	5.39 ± 8.21	4.90 ± 6.41	−0.49 ± 9.41	0.779	5.84 ± 6.82	4.61 ± 5.15	−1.23 ± 8.24	0.419	3.82 ± 6.84	3.02 ± 4.80	−0.80 ± 8.57	0.630	0.948
Vitamin C (mg)	115.22 ± 65.40	135.33 ± 68.58	20.11 ± 89.52	0.236	109.74 ± 87.04	102.07 ± 68.86	−6.68 ± 72.24	0.930	91.67 ± 51.49	118.41 ± 51.16	26.73 ± 58.88	0.026	0.635
Calcium (mg)	512.22 ± 147.42	499.36 ± 221.46	−12.85 ± 237.3	0.773	587.51 ± 346.72	522.00 ± 250.61	−65.51 ± 318.93	0.270	490.13 ± 247.10	547.06 ± 267.21	56.94 ± 316.30	0.358	0.635
Phosphorus (mg)	1072.28 ± 236.41	1006.02 ± 341.98	−66.26 ± 389.43	0.367	1009.47 ± 463.45	971.41 ± 336.96	−38.05 ± 378.95	0.587	869.79 ± 321.65	876.61. ± 320.15	6.86 ± 332.16-	0.915	0.635
MET (min/week)	2490.00 ± 2122.15	3002.14 ± 3423.55	512.14 ± 3697.90	0.470	3002.67 ± 3203.45	4350.00 ± 4649.10	1347.33 ± 4114.29	0.083	1822.67 ± 1418.57	4309.63 ± 5266.72	2486.96 ± 5241.23	0.021 *	0.251

Values are presented as mean ± SD. ^1^ Analyzed by paired *t*-test, ^2^ Analyzed by linear mixed model, * *p* < 0.05. Abbreviations: MET, metabolic equivalent.

**Table 4 nutrients-13-03352-t004:** Changes in bone turnover markers in subjects during the intervention period.

	GE Low (1 g) Group (*n* = 28)				GE High (3 g) Group (*n* = 30)				Placebo Group (*n* = 27)					
	0 Week	12 Week	Diff	*p*-Value ^1^	0 Week	12 Week	Diff	*p*-Value ^1^	0 Week	12 Week	Diff	*p*-Value ^1^	*p*-Value ^2^	*p*-Value ^3^
Osteocalcin (ng/mL)	21.30 ± 5.72	22.05 ± 5.77	0.74 ± 4.97 ^a^	0.436	18.93 ± 3.72	23.45 ± 5.81	4.52 ± 4.42 ^b^	<0.0001 ***	21.11 ± 6.34	21.50 ± 8.16	0.39 ± 4.31 ^a^	0.643	0.001 **	<0.001 ***
DPD (nM DPD/mM crea)	7.43 ± 1.33	6.86 ± 1.42	−0.58 ± 1.57	0.061	7.33 ± 1.52	7.23 ± 2.04	−0.11 ± 1.61	0.722	8.29 ± 4.74	7.85 ± 5.53	−0.44 ± 1.59	0.166	0.510	0.564
DPD/OC ratio	0.37 ± 0.10	0.33 ± 0.12	−0.04 ± 0.10 ^ab^	0.057	0.40 ± 0.09	0.32 ± 0.08	−0.08 ± 0.05 ^a^	<0.0001 ***	0.42 ± 0.26	0.41 ± 0.37	−0.01 ± 0.15 ^b^	0.772	0.046 *	0.027 *
CTX (ng/mL)	0.48 ± 0.11	0.48 ± 0.12	0.00 ± 0.08	0.802	0.49 ± 0.16	0.49 ± 0.14	0.00 ± 0.10	0.961	0.45 ± 0.16	0.47 ± 0.17	0.02 ± 0.11	0.411	0.683	0.918
OC/CTX ratio	44.48 ± 11.39	46.83 ± 11.08	2.32 ± 10.82 ^a^	0.268	41.10 ± 12.16	48.72 ± 11.51	7.49 ± 12.89 ^b^	0.004 **	49.92 ± 15.51	47.42 ± 11.78	−2.37 ± 12.54 ^a^	0.329	0.013 *	0.021 *
NTX (ng/mL)	15.58 ± 2.88	15.40 ± 2.58	−0.17 ± 1.63	0.580	15.95 ± 4.05	15.39 ± 3.74	−0.56 ± 2.17	0.170	15.93 ± 4.03	15.83 ± 4.42	−0.09 ± 2.37	0.837	0.664	0.757
Ca (mg/dL)	9.45 ± 0.36	9.42 ± 0.28	−0.03 ± 0.36 ^a^	0.679	9.35 ± 0.27	9.53 ± 0.32	0.18 ± 0.24 ^b^	<0.001 ***	9.33 ± 0.25	9.38 ± 0.25	0.06 ± 0.29 ^ab^	0.335	0.034 *	0.017 *
*p* (mg/dL)	3.67 ± 0.35	3.68 ± 0.37	0.00 ± 0.41	0.964	3.81 ± 0.33	3.72 ± 0.39	−0.08 ± 0.36	0.221	3.72 ± 0.49	3.81 ± 0.51	0.09 ± 0.43	0.274	0.263	0.173
BsALP(μg/L)	16.59 ± 5.06	16.31 ± 3.67	−0.28 ± 4.05	0.719	15.91 ± 3.56	15.07 ± 3.63	−0.84 ± 2.46	0.071	17.23 ± 5.93	17.07 ± 5.84	−0.16 ± 2.87	0.771	0.689	0.720
P1NP (pg/mL)	61.23 ± 13.69	58.46 ± 12.98	−2.77 ± 6.63	0.036*	59.10 ± 14.97	59.01 ± 16.15	−0.09 ± 7.31	0.947	58.18 ± 15.98	61.29 ± 21.89	3.11 ± 13.91	0.256	0.087	0.170

Values are presented as mean ± SD. ^1^ Analyzed by paired *t*-test. ^2^ Analyzed by linear mixed model. ^3^ RM-ANCOVA after adjustment for the effects of compliance and MET value. * *p* < 0.05, ** *p* < 0.01, *** *p* < 0.001 Multiple comparison by Bonferroni correction. Different letters indicate statistically significant differences between groups. Abbreviations: MET, metabolic equivalent, DPD, deoxypyridinoline; CTX, C-terminal telopeptide, NTX, N-terminal telopeptide, BsALP, bone-specific alkaline phosphatase, P1NP, procollagen type 1 n-terminal propeptide, WOMAC, Western Ontario and McMaster Universities Arthritis.

**Table 5 nutrients-13-03352-t005:** Changes in WOMAC scores in subjects during the intervention period.

	GE Low (1 g) Group (*n* = 28)				GE High (3 g) Group (*n* = 30)				Placebo Group (*n* = 27)					
	0 Week	12 Week	Diff	*p*-Value ^1^	0 Week	12 Week	Diff	*p*-Value ^1^	0 Week	12 Week	Diff	*p*-Value ^1^	*p*-Value ^2^	*p*-Value ^3^
**WOMAC index**														
Pain														
Q1	0.36 ± 0.68	0.57 ± 0.96	0.21 ± 0.74	0.136	0.50 ± 0.68	0.40 ± 0.56	−0.10 ± 0.71	0.448	0.22 ± 0.42	0.26 ± 0.53	0.04 ± 0.52	0.713	0.205	0.184
Q2	0.50 ± 0.69	0.68 ± 0.77	0.18 ± 0.67	0.170	0.87 ± 1.01	0.77 ± 0.73	−0.10 ± 0.92	0.558	0.48 ± 0.51	0.63 ± 0.63	0.15 ± 0.66	0.256	0.319	0.248
Q3	0.18 ± 0.39	0.21 ± 0.57	0.04 ± 0.58 ^ab^	0.746	0.57 ± 0.97	0.30 ± 0.65	−0.27 ± 0.98 ^a^	0.147	0.11 ± 0.42	0.33 ± 0.62	0.22 ± 0.42 ^b^	0.011 *	0.037 *	0.122
Q4	0.18 ± 0.39	0.36 ± 0.62	0.18 ± 0.55	0.096	0.33 ± 0.76	0.30 ± 0.47	−0.03 ± 0.85	0.832	0.30 ± 0.61	0.37 ± 0.69	0.07 ± 0.62	0.537	0.507	0.459
Q5	0.18 ± 0.39	0.29 ± 0.60	0.11 ± 0.50	0.264	0.33 ± 0.66	0.40 ± 0.62	0.07 ± 0.74	0.625	0.19 ± 0.48	0.37 ± 0.69	0.19 ± 0.74	0.202	0.796	0.740
Stiffness														
Q1	0.46 ± 0.74	0.64 ± 0.78	0.18 ± 0.61	0.134	0.73 ± 0.78	0.60 ± 0.81	−0.13 ± 1.01	0.475	0.33 ± 0.55	0.44 ± 0.75	0.11 ± 0.64	0.376	0.283	0.249
Q2	0.46 ± 0.69	0.64 ± 0.91	0.18 ± 0.82^a^	0.259	1.03 ± 0.81	0.63 ± 0.72	−0.40 ± 0.77 ^b^	0.008 **	0.48 ± 0.70	0.52 ± 0.70	0.04 ± 0.76 ^a^	0.802	0.016 *	0.011^*^
Physical function														
Q1	0.64 ± 0.91	0.75 ± 0.93	0.11 ± 0.79	0.477	0.67 ± 0.88	0.73 ± 0.87	0.07 ± 0.91	0.690	0.59 ± 0.64	0.52 ± 0.58	−0.07 ± 0.62	0.537	0.669	0.892
Q2	0.68 ± 0.82	0.68 ± 0.82	0.00 ± 0.67	>0.999	0.77 ± 0.86	0.63 ± 0.76	−0.13 ± 0.68	0.293	0.52 ± 0.58	0.59 ± 0.64	0.07 ± 0.62	0.537	0.482	0.296
Q3	0.39 ± 0.79	0.61 ± 0.83	0.21 ± 0.79	0.161	0.77 ± 0.86	0.67 ± 0.80	−0.10 ± 1.06	0.610	0.3 ± 0.55	0.63 ± 0.79	0.30 ± 0.78	0.058	0.209	0.320
Q4	0.46 ± 0.74	0.50 ± 0.69	0.04 ± 0.58	0.746	0.57 ± 0.73	0.53 ± 0.86	−0.03 ± 1.07	0.865	0.37 ± 0.56	0.41 ± 0.75	0.04 ± 0.76	0.802	0.934	0.801
Q5	0.54 ± 0.74	0.46 ± 0.58	−0.07 ± 0.81	0.646	0.63 ± 0.85	0.53 ± 0.68	−0.10 ± 0.99	0.586	0.30 ± 0.67	0.37 ± 0.79	0.07 ± 0.92	0.678	0.749	0.687
Q6	0.43 ± 0.74	0.50 ± 0.75	0.07 ± 0.72	0.602	0.67 ± 0.88	0.47 ± 0.73	−0.20 ± 1.06	0.312	0.30 ± 0.47	0.44 ± 0.70	0.15 ± 0.66	0.256	0.261	0.127
Q7	0.50 ± 0.75	0.50 ± 0.75	0.00 ± 0.47	>0.999	0.53 ± 0.73	0.60 ± 0.77	0.07 ± 0.87	0.677	0.33 ± 0.55	0.52 ± 0.75	0.19 ± 0.62	0.134	0.596	0.715
Q8	0.32 ± 0.72	0.54 ± 0.84	0.21 ± 0.63	0.083	0.50 ± 0.82	0.6 ± 0.81	0.10 ± 0.84	0.522	0.26 ± 0.45	0.26 ± 0.53	0.00 ± 0.55	>0.999	0.521	0.602
Q9	0.32 ± 0.61	0.39 ± 0.79	0.07 ± 0.54	0.490	0.33 ± 0.55	0.33 ± 0.55	0.00 ± 0.69	>0.999	0.19 ± 0.40	0.37 ± 0.69	0.19 ± 0.56	0.096	0.512	0.437
Q10	0.61± 0.79	0.46 ± 0.74	−0.14 ± 0.65	0.255	0.63 ± 0.76	0.53 ± 0.57	−0.10 ± 0.80	0.501	0.22 ± 0.51	0.48 ± 0.85	0.26 ± 0.66	0.050	0.076	0.106
Q11	0.25 ± 0.59	0.43 ± 0.79	0.18 ± 0.55	0.096	0.33 ± 0.55	0.37 ± 0.61	0.03 ± 0.72	0.801	0.19 ± 0.40	0.41 ± 0.8	0.22 ± 0.64	0.083	0.506	0.541
Q12	0.29 ± 0.71	0.39 ± 0.74	0.11 ± 0.57	0.326	0.50 ± 0.68	0.43 ± 0.57	−0.07 ± 0.78	0.645	0.15 ± 0.36	0.37 ± 0.79	0.22 ± 0.64	0.083	0.269	0.212
Q13	0.29 ± 0.60	0.39 ± 0.69	0.11 ± 0.50	0.264	0.43 ± 0.77	0.43 ± 0.73	0.00 ± 0.98	>0.999	0.30 ± 0.47	0.41 ± 0.75	0.11 ± 0.58	0.327	0.804	0.751
Q14	0.32 ± 0.55	0.43 ± 0.79	0.11 ± 0.69	0.415	0.53 ± 0.86	0.57 ± 0.86	0.03 ± 0.93	0.845	0.22 ± 0.51	0.44 ± 0.64	0.22 ± 0.58	0.056	0.637	0.699
Q15	0.32 ± 0.61	0.36 ± 0.68	0.04 ± 0.64	0.769	0.27 ± 0.52	0.33 ± 0.76	0.07 ± 0.87	0.677	0.26 ± 0.53	0.48 ± 0.75	0.22 ± 0.80	0.161	0.636	0.810
Q16	1.04 ± 0.92	1.04 ± 0.96	0.00 ± 0.82	>0.999	1.23 ± 0.94	1.20 ± 0.76	−0.03 ± 1.03	0.861	0.70 ± 0.91	0.96 ± 0.94	0.26 ± 0.81	0.110	0.418	0.567
Q17	0.82 ± 0.82	0.68 ± 0.67	−0.14 ± 0.97	0.443	0.97± 0.89	0.77 ± 0.86	−0.20 ± 0.76	0.161	0.67 ± 0.83	0.63 ± 0.74	−0.04 ± 0.85	0.824	0.773	0.730
Total score	10.54±12.75	12.50 ± 13.91	1.96 ± 7.96	0.202	14.70 ± 12.63	13.13 ± 12.14	−1.57 ± 12.71	0.505	8.00 ± 9.12	11.22 ± 13.69	3.22 ± 9.88	0.102	0.202	0.162

Values are presented as mean ± SD. ^1^ Analyzed by paired *t*-test. ^2^ Analyzed by linear mixed model, ^3^ Analyzed by linear mixed model after adjustment for the effects of compliance and MET value. * *p* < 0.05, ** *p* < 0.01. Multiple comparison by Bonferroni correction. Different letters indicate statistically significant differences between groups. Abbreviations: MET, metabolic equivalent, Pain part Q1: descending stair; Q2: stair climbing; Q3: when sleep at night; Q4: sitting; Q5: standing upright. Stiffness part Q1: morning stiffness; Q2: stiffness occurring later in the day. Physical function part Q1: descending stair; Q2: ascending stair; Q3: rising from sitting; Q4: standing, Q5: bending to floor; Q6: walking on flat; Q7: getting in/out of car; Q8: going shopping; Q9: putting on socks; Q10: rising from bed; Q11: taking off socks; Q12: lying from bed; Q13: getting in/out of bath; Q14: sitting; Q15: getting on/off toilet; Q16: heavy domestic duties; Q17: light domestic in the day.

## Data Availability

The datasets generated during and/or analyzed during the current study are available from the corresponding author on reasonable request.

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
