# Peer review of "Effect of Ginseng Extracts on the Improvement of Osteopathic and Arthritis Symptoms in Women with Osteopenia: A Randomized, Double-Blind, Placebo-Controlled Clinical Trial"

_nutrients, 2021, doi:10.3390/nu13103352_

Round 1

Reviewer 1 Report

12 weeks of study osteoporosis is not enough to assess osteoporosis .

OC/CTX , can not indicate effectivness of the treatment .

Can not do DEXA in this short period

Need longer trials , fracture risk, DEXA results 

Author Response

Dear Reviewer 1,

Q1. 12 weeks of study osteoporosis is not enough to assess osteoporosis.

We agree with the reviewer’s comment, and we have mentioned it as one of the limitations of the study. However, in this short-term trial, the major results of the study are consistent with those of the previous in vitro and in vivo observations.  This study is the first randomized, double-blind, placebo-controlled clinical trial to determine the effect of GE on improving bone metabolism and arthritis symptoms in postmenopausal women with osteopenia. We hope that observations of this trial will draw further scientific attention.

Q2. OC/CTX, cannot indicate effectiveness of the treatment.

We appreciate the reviewer’s valuable suggestions. We agree that OC/CTX does not indicate the effectiveness directly but is used as a marker to evaluate the potential effects. In this study, we have considered OC/CTX as secondary measures. As suggested, we have revised the content and corrected the sentences that suggested the observations of OC/CTX indicate the effectiveness of the treatment.

Q3. Cannot do DEXA in this short period

We completely agree with the reviewer’s opinion. Here, we used DEXA to screen volunteers (during the selection process). For efficacy evaluation, bone formation index was evaluated, not bone density.

Q4. Need longer trials, fracture risk, DEXA results

On the length of the trial, we do have similar views. We have mentioned it as one of the limitations of the study. We will consider fracture risk and DEXA observations in our future long-term clinical trials.

Reviewer 1.

Q1. 12 weeks of study osteoporosis is not enough to assess osteoporosis.

We agree with the reviewer’s comment, and we have mentioned it as one of the limitations of the study. However, in this short-term trial, the major results of the study are consistent with those of the previous in vitro and in vivo observations.  This study is the first randomized, double-blind, placebo-controlled clinical trial to determine the effect of GE on improving bone metabolism and arthritis symptoms in postmenopausal women with osteopenia. We hope that observations of this trial will draw further scientific attention.

Q2. OC/CTX, cannot indicate effectiveness of the treatment.

We appreciate the reviewer’s valuable suggestions. We agree that OC/CTX does not indicate the effectiveness directly but is used as a marker to evaluate the potential effects. In this study, we have considered OC/CTX as secondary measures. As suggested, we have revised the content and corrected the sentences that suggested the observations of OC/CTX indicate the effectiveness of the treatment.

Q3. Cannot do DEXA in this short period

We completely agree with the reviewer’s opinion. Here, we used DEXA to screen volunteers (during the selection process). For efficacy evaluation, bone formation index was evaluated, not bone density.

Q4. Need longer trials, fracture risk, DEXA results

On the length of the trial, we do have similar views. We have mentioned it as one of the limitations of the study. We will consider fracture risk and DEXA observations in our future long-term clinical trials.

Reviewer 2 Report

The manuscript entitled “Effect of ginseng extracts on the improvement of osteopathic ad arthritis symptoms in women with osteopenia: A randomized, double-blind, placebo-controlled clinical trial” by Su-Jin Jung et al., aims to show the relevance of Ginseng extract (GE) in woman with post menopausal osteopenia

Overall, the paper is well written, very descriptive and full of information and results. The materials and methods and results, are extensively illustrated. In general, the paper gives interesting information on the possible relationship among osteoporosis markers and GE administration in this randomized, double-blind, placebo-controlled trial.

Nevertheless, I would like the authors comment on the following issues:

  • How to analyze the amount of GE absorbed in the body? Is the extract found in blood or urine?
  • Since diet Ca, vitamin D and protein largely influence bone density in adult it might have been more powerful if you would have given a controlled diet during these 12 weeks, could you comment on the feasibility of it?
  • Glucose and insulin are proven to influence bone metabolism, do you have any details on these parameters?
  • are strictly linked to Several studies, including human studies, have indicated that Gut hormones (GLP-1, GLP-2, GIP) enhance bone formation (doi: 10.3389/fendo.2019.00075.), do you have any information on your group of people?

I overall, think that the paper is informative and gives interesting suggestions on osteoporosis and food integration (GE integration) although a longer period of administration is suggested

Author Response

Dear Reviewer 2, 

Q1. How to analyze the amount of GE absorbed in the body? Is the extract found in blood or urine?
We thank the reviewer for the valuable comment. In this study, the amount of GE absorbed was not investigated. Generally, ginger phenolics occurred as phase II metabolites in plasma, and the pharmacokinetic of GE is mainly measured by human plasma.

Reference

- Yun-Jung Kim et al., Validation of LC-MS/MS method for determination of ginsenoside Rg1 in human plasma. ANALYTICALSCIENCE&TECHNOLOGY 2013, 26(4): 221-227.

- J.Y Jeon Et al., An assessment of the Pharmacokinetics of DW1029M in healthy Korean Subjects. Clinical therapeutics, 2015.

Q2. Since diet Ca, vitamin D and protein largely influence bone density in adult it might have been more powerful if you would have given a controlled diet during these 12 weeks, could you comment on the feasibility of it?

We appreciate the reviewer’s comments on the matter. Yes, a controlled diet with Ca and vitamin D would have added value to the trial. However, we aim to evaluate the effectiveness of GE with a normal/regular diet. Thus, we asked participants not to consume any other functional foods or dietary supplements during the study period. Also, study participants were asked to record the dietary intakes for 3 days (two days of the week, one day of the weekend) before the first visit (week 0) and the third visit (week 12). Based on these recorded and retrieved diet records, average daily dietary intake was analyzed. Further, dietary intake data were analyzed by a professional nutritionist. These analyses suggested that Ca and Vitamin D intake during the study period remained similar in all the groups. Thus, we are assured of feasibility of the trial. Considering the comment, discussion on the same is included in the revised manuscript. Updated text as follows,  

“In middle-aged women, calcium supplements have been reported to contribute to significant increases in blood OC and bone density [15-17]. In this study, dietary intake analysis revealed that vitamin D, calcium, and protein intake during participation are similar in two groups. Thus, it can be presumed that these dietary intakes do not influence the outcome of the study”

Q3. Glucose and insulin are proven to influence bone metabolism, do you have any details on these parameters?

In this study, blood glucose was not shown as an evaluation index. However, glucose levels were evaluated as the safety evaluation index. During the study period, there was no significant difference in glucose levels between the groups. We presented the blood glucose index as a safety evaluation index (Supplementary Table 1).

Q4. Are strictly linked to several studies, including human studies, have indicated that Gut hormones (GLP-1, GLP-2, GIP) enhance bone formation (doi: 10.3389/fendo.2019.00075.), do you have any information on your group of people?

We thank the reviewers for sharing the potential importance of gut hormones in bone formation. Unfortunately, we have not evaluated gut hormones in this study. We will consider evaluating gut hormones in our future clinical trials relevant to bone health.

Q5. I overall, think that the paper is informative and gives interesting suggestions on osteoporosis and food integration (GE integration) although a longer period of administration is suggested

On length of the trial, we do have similar views. We have mentioned it as one of the limitations of the study.
